# Enhancing Healthcare Integrity Using Simple Statistical Methods: Detecting Irregularities in Historical Dermatology Services Payments

**DOI:** 10.3390/healthcare13121464

**Published:** 2025-06-18

**Authors:** Andrej F. Plesničar, Nena Bagari Bizjak, Pika Jazbinšek

**Affiliations:** Health Insurance Institute of Slovenia, SI 1000 Ljubljana, Slovenia; nena.bagari-bizjak@zzzs.si (N.B.B.); pika.jazbinsek@zzzs.si (P.J.)

**Keywords:** healthcare service payments, claims data, data credibility, fraud detection, outlier detection, Benford’s Law, Grubbs’ test, Hampel’s test, T-test, dermatology

## Abstract

**Background and Objectives:** Healthcare payment systems face challenges such as fraud and overbilling, which often require costly and resource-intensive detection tools. In response, the utility of simple statistical tests was explored in this study as a practical alternative for identifying irregularities in dermatology service payments within the Health Insurance Institute of Slovenia (HIIS). **Materials and Methods:** Ten-year-old anonymized billing data from 30 dermatology providers in Slovenia (with a population of 2 million) were analyzed to evaluate the effectiveness of the proposed methodology while aiming to avoid reputational harm to current providers. The dataset from 2014 included variables such as the “number of services charged”, “total number of points charged” (under Slovenia’s point-based tariff system at the time), “number of points per examination”, “average examination values (EUR)”, “number of first examinations”, and “total number of first/follow-up examinations”. Data credibility was assessed using Benford’s Law (for calculating χ^2^ values and testing null hypothesis rejection at the 95% level), and Grubbs’ test, Hampel’s test, and T-test were used to identify outliers. **Results:** An analysis using Benford’s Law revealed significant deviations for the “number of services charged” (*p* < 0.005), “total number of points charged” (*p* < 0.01), “number of points per examination” (*p* < 0.0005), and “average examination values (EUR)” (*p* < 0.005), suggesting anomalies. Conversely, data on the numbers of “first” (*p* < 0.7) and “total first/follow-up examinations” (*p* < 0.3) were found to align with Benford’s Law, indicating authenticity. Outlier detection consistently identified two institutions with unusually high values for points per examination and average examination monetary value. **Conclusions:** Simple statistical tests can effectively identify potential irregularities in healthcare payment data, providing a cost-effective screening method for further investigation. Identifying outlier providers highlights areas needing detailed scrutiny to understand anomaly causes.

## 1. Introduction

Healthcare systems worldwide face significant challenges such as fraud, overbilling, and waste, which can lead to substantial financial losses and compromise the quality of patient care [1,2]. Detecting such irregularities is crucial for maintaining the integrity and efficiency of healthcare services. However, modern fraud detection methods used in developed countries often rely on complex and resource-intensive technologies, which can be costly and inaccessible to many healthcare systems [2,3,4]. These advanced technologies require substantial resources and infrastructure due to their cost and complexity, further limiting their accessibility. In contrast, some practical and simple statistical tests may offer a cost-effective alternative for identifying potential anomalies in healthcare payment data.

Healthcare payment integrity remains a cornerstone of ethical service delivery, requiring vigilant oversight to maintain public trust and system sustainability [5]. While advanced analytics dominate modern fraud detection, resource-constrained systems often lack access to these technologies, creating vulnerabilities. This challenge mirrors broader ethical tensions in healthcare financing, where conflicts of interest and data complexity can undermine transparency [6,7]. Recent studies have emphasized the growing need for accessible tools, particularly as healthcare systems grapple with big data integration challenges and evolving payment models [8,9]. Our study addresses this gap by evaluating simple statistical methods—Benford’s Law and outlier detection tests—as pragmatic alternatives for identifying irregularities in dermatology billing data. This approach aligns with calls for cost-effective solutions that balance technological limitations with the ethical imperative of financial accountability [5,7], while providing actionable insights for healthcare leadership navigating integrity preservation [8].

Benford’s Law is particularly useful for assessing data credibility and authenticity by identifying deviations in the expected frequency distribution of leading digits in datasets. This method can act as a first-line tool for detecting potential fraud by highlighting data that do not conform to natural patterns [10,11,12]. Additionally, outlier detection methods such as Grubbs’ test, Hampel’s test, and T-test are designed to identify extreme values in datasets. These tests help pinpoint data points that significantly deviate from the norm, which may indicate anomalies or irregularities in billing practices [13].

Despite their potential, simple statistical methods, including Benford’s Law and some of the outlier detection tests, have been underutilized in healthcare fraud detection [13]. These methods may provide quick insights into large datasets and can serve as initial screening tools to identify potential irregularities. By leveraging these simple yet effective tools, healthcare systems could enhance their ability to detect anomalies without relying on expensive technologies, ultimately improving the efficiency and integrity of healthcare services.

This study evaluated the practicality and usefulness of Benford’s Law and simple statistical tests, including Grubbs’ test, Hampel’s, test and T-test, for detecting irregularities in dermatology service payments within the HIIS. The decision to use anonymized historical billing data from 2014 was strategic, as it provided a controlled environment for testing these methods without risking reputational harm to current providers. By analyzing historical data, this study enabled a retrospective evaluation of fraud detection techniques, offering insights into how fraud patterns may have evolved over time. This approach not only informs future strategies for improving healthcare payment systems but also highlights the potential of these methods as cost-effective initial screening tools for identifying anomalies and billing errors.

This study contributes to the academic literature in three keyways. First, it demonstrates the practical application of established statistical methods in a novel context—healthcare payment integrity—bridging theoretical foundations with real-world applications. Second, it provides empirical evidence for the effectiveness of simple yet powerful analytical tools in resource-constrained healthcare systems. Third, it offers a methodological framework for preliminary fraud detection that can be adapted across various healthcare specialties, potentially influencing both policy and practice in healthcare management.

Slovenia’s healthcare system operates under a mandatory social health insurance (SHI) model, with the HIIS as the single payer. Services are reimbursed through a hybrid system combining fee-for-service (FFS), capitation, and diagnosis-related groups (DRGs) [14]. We selected anonymized billing data from 2014 to mark the dataset’s tenth anniversary as this study started in late 2024, providing a meaningful retrospective period. This choice ensured a stable pre-reform context, minimized reputational risk to current providers, and avoided confounding effects from recent events such as the COVID-19 pandemic. Dermatology was chosen as the focus because it features a diverse mix of diagnostic and procedural services, resulting in variable billing patterns, and offers a manageable dataset size for comprehensive analysis. Additionally, its billing structure is representative of broader healthcare payment systems in Slovenia. While this study focuses on dermatology, the statistical methods applied—digit distribution analysis and outlier detection—are broadly adaptable to other medical specialties and payment systems, as their underlying principles remain consistent across contexts.

The main conclusions of this study highlight the potential of simple statistical tests to identify outliers and anomalies in healthcare payment data, providing a potential practical and useful approach for healthcare systems with limited resources.

## 2. Materials and Methods

The dataset analyzed in this study consisted of billing data submitted to the HIIS in 2014 by 30 dermatology providers operating in Slovenia during that year. The dataset included variables such as the “number of services charged”, “total number of points charged” (calculated under Slovenia’s point-based tariff system at the time), “number of points per examination”, “average examination values in euros (EUR)”, “number of first examinations”, and “total number of first and follow-up examinations”. The data were collected from existing records and anonymized to ensure compliance with ethical standards and protect provider confidentiality. This comprehensive dataset provided a robust foundation for applying statistical methods to detect anomalies and assess billing practices.

Benford’s Law was used to evaluate the distribution of leading digits in the dataset. This method predicts that the leading digits in naturally occurring datasets should follow a specific logarithmic distribution [10,11,12]. The analysis was conducted using the Miller–Nigrini Excel Benford Tester [15], which calculates the observed probability of each leading digit and compares it with the expected probability according to Benford’s Law. The chi-square (χ^2^) test was applied to determine if the observed distribution significantly deviated from the expected distribution at 95% confidence levels. The *p*-values from Benford’s Law analysis determined whether the observed data distributions aligned with the expected logarithmic pattern. A *p*-value greater than 0.05 indicated alignment with Benford’s Law, while a *p*-value less than 0.05 signaled statistically significant deviations from the expected distribution.

Grubbs’ test was employed to detect outliers in the dataset by identifying data points that significantly deviate from the mean. This method assumes that data are normally distributed and is particularly suitable for small- to moderate-sized datasets. Real-world billing data often exhibit right-skewed distributions; however, Grubbs’ test is robust to mild deviations when identifying extreme outliers in administrative datasets. Formal normality tests were omitted because the HIIS data structures were well characterized in previous audits, and our goal was pragmatic anomaly detection, not parametric modeling. Grubbs’ test measures how far a suspected outlier lies from the mean in terms of standard deviation. The calculated GG-value is then compared to a critical threshold derived from the t-distribution to determine whether the data point is statistically significant at a specified confidence level (α = 0.05). The simplicity of Grubbs’ test makes it widely applicable in various fields, including healthcare fraud detection, where identifying anomalies in payment data can reveal irregularities or billing errors [13,16,17].

Hampel’s test is a statistical method for outlier detection that relies on the median and median absolute deviation (MAD) instead of the mean and standard deviation, and it is effective for datasets that do not follow a normal distribution or contain skewed data. The test involves calculation of the median, which is used as a measure of central tendency, because the median is less sensitive to outliers compared to the mean; it then computes the MAD and determines a threshold multiplier. A commonly used threshold is *k* = 4.5, which corresponds to approximately three standard deviations in a normal distribution. A data point is thus considered an outlier if it exceeds 4.5 times the MAD from the median [13,16,18].

The T-test for outlier detection was used to identify whether a suspected outlier significantly distorts the distribution of the dataset. This approach assumes that the data under examination are approximately normally distributed. The procedure begins by calculating the mean and standard deviation of the full dataset including the suspected outlier. After excluding the outlier, the mean and standard deviation are recalculated for the trimmed dataset. The deviation between the suspected outlier and the trimmed mean is then computed. This deviation is compared to a threshold derived from a T-test reference table, which accounts for the sample size and desired confidence level. If the deviation exceeds this threshold, the data point is considered a statistically significant outlier [13,19,20].

The statistical analyses for detecting outliers in the dataset described above were conducted using Excel 2010, which provides an accessible platform for performing calculations and verifying results. Before applying these tests, the dataset was carefully examined for missing values and inconsistencies to ensure the reliability of the analyses.

## 3. Results

This study analyzed dermatology service payments using Benford’s Law and outlier detection methods. The results show significant deviations from expected distributions for certain billing categories, indicating potential anomalies. Outlier detection consistently identified certain providers with unusually high values for different billing categories.

### 3.1. Benford’s Law Analysis

The application of Benford’s Law to the dataset revealed varying levels of compliance across different categories of data. The statistical results are summarized in Table 1.

“Number of services charged”: The observed distribution of leading digits in this category significantly deviated from the expected Benford distribution (Figure 1 and Table 1). This deviation suggests potential anomalies in the billing process.
Figure 1Leading digit test: the comparison between the observed probability and the Benford probability of the frequency distribution of leading digits in the “number of services charged” data.
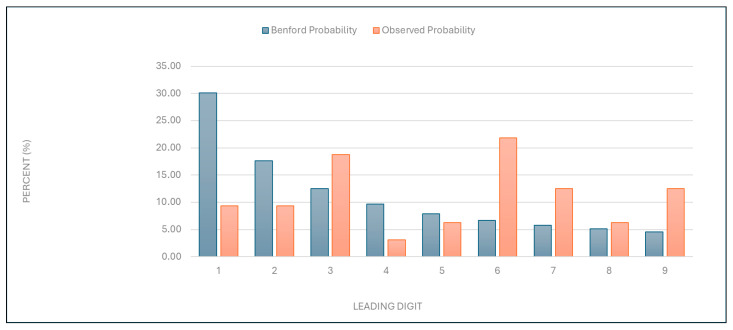

“Total number of points charged”: Similarly to the results described above, the data for this category were found to deviate from the expected distribution (Figure 2 and Table 1).
Figure 2Leading digit test: the comparison between the observed probability and the Benford probability of the frequency distribution of leading digits in the “total number of points charged” data.
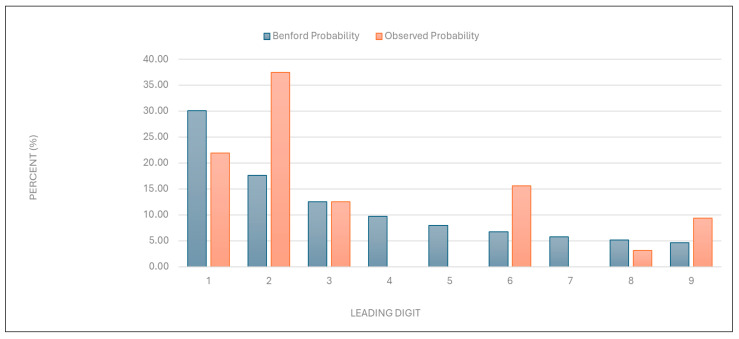

“Number of points per examination”: The data were found to deviate significantly from the expected distribution (Figure 3 and Table 1). This suggests potential overbilling or anomalies in service evaluation.
Figure 3Leading digit test: the comparison between the observed probability and the Benford probability of the frequency distribution of leading digits in the “number of points per examination” data.
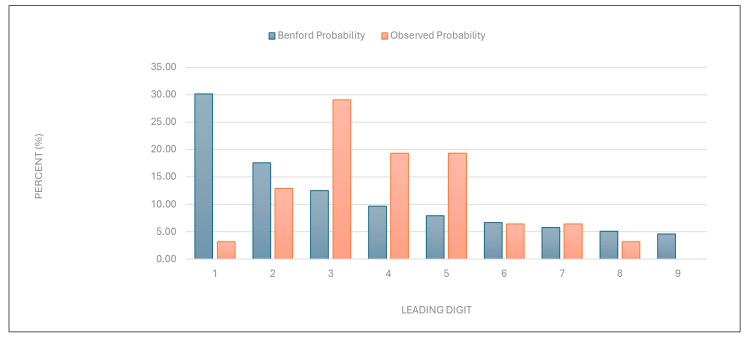

“Average examination value (EUR)”: The data for this category were also found to deviate significantly from the expected distribution (Figure 4 and Table 1), indicating potential irregularities in examination pricing.
Figure 4Leading digit test: the comparison between the observed probability and the Benford probability of the frequency distribution of leading digits in the “average examination value (EUR)” data.
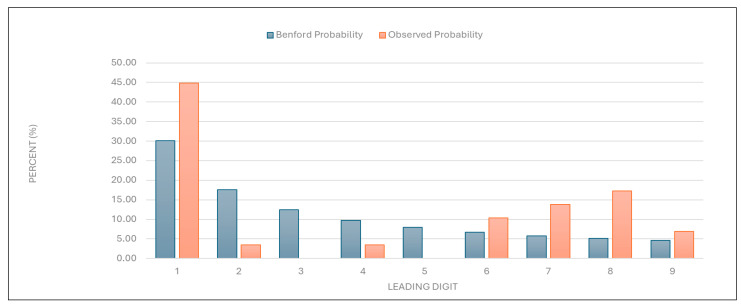

“Number of first examinations”: The data for this category were found to align with the expected Benford distribution (Figure 5 and Table 1). This result suggests that the data on the number of first examinations are likely authentic.
Figure 5Leading digit test: the comparison between the observed probability and the Benford probability of the frequency distribution of leading digits in the “number of first examinations” data.
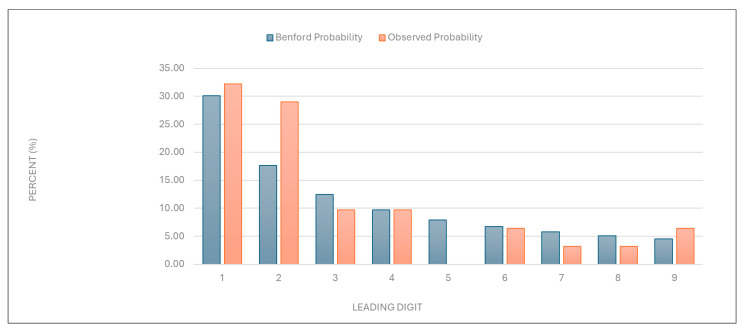

“Total number of first and follow-up examinations”: The data for this category were found to align with the expected distribution (Figure 6 and Table 1). This result suggests that the total examination numbers are likely authentic.
Figure 6Leading digit test: the comparison between the observed probability and the Benford probability of the frequency distribution of leading digits in the “total number of first and follow-up examinations” data.
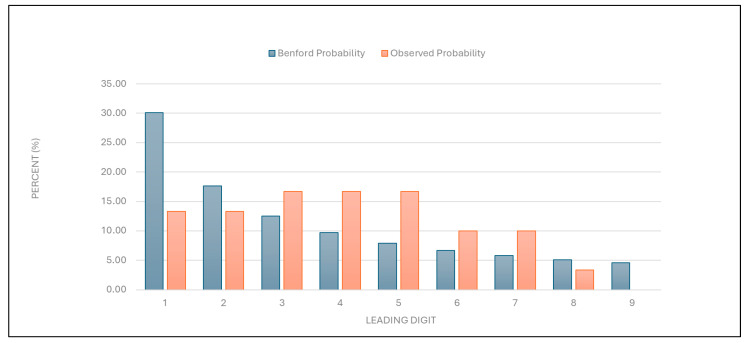


### 3.2. Outlier Detection

Outlier detection was performed using Grubbs’ test, Hampel’s test, and T-test to identify providers with unusually high values for various billing categories.

#### 3.2.1. Grubbs’ Test

“Number of services charged”: One of the two large university health centers in the country was identified as an outlier, with a GG-value of 4.906761. This result suggests that its billing practices differ significantly from the norm, possibly due to factors such as its size and patient demographics, with a larger number of elderly patients, or its unique billing practices.“Total number of points charged”: The same large university healthcare center as mentioned above was identified as an outlier, with a GG-value of 4.955756, indicating unusual point billing practices.“Number of points per examination”: A specialized, privately owned dermatology center was identified as an outlier, with a GG-value of 4.494556, indicating an unusually high number of points per examination.“Average examination value (EUR)”: The same specialized dermatology center was identified as an outlier, with a GG-value of 6.243344, suggesting unusually high examination values. Another dermatology clinic within a primary healthcare center in a major city was also identified as an outlier, with a GG-value of 3.062068.“Number of first examinations”: The same specialized dermatology center was again identified as an outlier, with a GG-value of 4.680405, suggesting an unusually high number of first examinations.“Total number of examinations”: The same specialized dermatology center was identified as an outlier, with a GG-value of 4.458205, suggesting an unusually high number of total examinations with a larger number of follow-up examinations.

#### 3.2.2. Hampel’s Test

“Number of services charged” and “total number of points charged”: No outliers were detected using Hampel’s test for these categories.“Number of points per examination”: A specialized, privately owned dermatology center and a primary healthcare center in a major city, both mentioned above, were identified as outliers. This result suggests that these centers charge an unusually high number of points per examination.“Average examination value (EUR)”: The same specialized, privately owned dermatology center was identified as an outlier, while the primary healthcare center in a major city was identified as being close to the outlier threshold.“Number of first examinations”: The same large university health center mentioned above was identified as an outlier. This suggests that the center’s number of first examinations was unusually high compared to other providers.“Total number of examinations”: The same large university health center was again identified as an outlier. This suggests that the center has a high number of follow-up examinations, possibly due to a larger number of elderly patients or its unique billing practices.

#### 3.2.3. T-Test

“Number of services charged”: The same large university health center was identified as an outlier. This again suggests that the center’s service billing practices differ significantly from the norm.“Total number of points charged”: No outliers were detected.“Number of points per examination”: The specialized, privately owned dermatology center mentioned above was identified as an outlier. This again indicates that the center charges an unusually high number of points per examination.“Average examination value (EUR)”: The same specialized, privately owned dermatology center was identified as an outlier. This suggests that the center’s average examination value is significantly higher than expected.“Number of first examinations”: The large university health center mentioned above was again identified as an outlier. This indicates that its number of first examinations is unusually high.“Total number of examinations”: The same large university health center was identified as an outlier. This suggests that the center also has an unusually high number of follow-up examinations.

## 4. Discussion

This study evaluated the practicality and usefulness of Benford’s Law and outlier detection tests (Grubbs’ test, Hampel’s test, and T-test) for identifying irregularities in healthcare billing data. Benford’s Law demonstrated significant utility in flagging deviations from expected digit distributions, signaling potential anomalies and casting doubt on data credibility and authenticity. The outlier detection tests complemented this analysis by pinpointing specific providers with anomalous billing patterns.

The accessibility of these methods makes them practical tools for healthcare systems with limited resources, enabling the rapid identification of irregularities in large datasets without the need for complex analytical infrastructure. Benford’s Law can serve as a first-line screening tool, while the outlier detection tests additionally provide a clear focus for further investigation by isolating extreme values that may indicate fraudulent or irregular practices.

This systematic approach highlights the complementary strengths of these methods, whose application can enhance transparency in healthcare systems and enable the early detection of anomalies. The simplicity and practicality of these tools make them especially valuable for resource-constrained settings, where advanced analytical expertise may not be readily available. By combining Benford’s Law with outlier detection tests, this study offers a cost-effective strategy for enhancing transparency in healthcare systems and informing targeted approaches to mitigate financial losses.

### 4.1. Strategic Use of Historical Data for Evaluating Fraud Detection Methods in Healthcare Billing

This study strategically utilized historical billing data from 2014 to evaluate the effectiveness of Benford’s Law as well as Grubbs’ test, Hampel’s test, and T-test in identifying anomalies within healthcare billing practices. As detailed in the Introduction, the 2014 dataset provided a stable pre-reform baseline while minimizing reputational risk for current providers. By focusing on historical real-world data, this study ensured that its primary objective was a methodological evaluation rather than the identification of present-day fraudsters.

Evaluating historical data may be particularly relevant in Slovenia, where healthcare financing relies on a combination of fee-for-service, capitation, and DRG reimbursement models, all of which are influenced by historical expenditure patterns [20,21,22]. Additionally, historical datasets often feature suitable sample sizes and fewer biases, such as nonresponse or incomplete records, which can compromise the validity of analyses [23,24,25,26]. These characteristics made the 2014 dataset an ideal dataset for evaluating the applicability and reliability of Benford’s Law and outlier detection tests in healthcare billing systems [21,22].

### 4.2. Assessing Data Credibility and Authenticity with Benford’s Law: Practicality and Usefulness in Healthcare Fraud Detection

Benford’s Law demonstrated its utility as a preliminary screening tool for checking the credibility and authenticity of the data, suggesting potential manipulation or overbilling. It detected anomalies in healthcare billing data for most of the variables examined in this study, such as “number of services charged”, “total number of points charged”, “number of points per examination”, and “average examination values (EUR)”. Conversely, compliance with Benford’s Law for categories such as “number of first examinations” and “total number of examinations” indicates that these data are likely authentic, reinforcing the method’s reliability for distinguishing between genuine and anomalous patterns. The deviations from expected digit distributions in these variables signaled potential irregularities, consistent with prior studies applying Benford’s Law to fraud detection. Its application has already proven effective in identifying billing anomalies and manipulated data in other studies [1,2,3,9,10,11,12,13]. Variables such as “number of first examinations” exhibited natural digit distributions due to their inherent constraints in ambulatory settings—patients typically have one initial visit per condition, limiting opportunities for manipulation. Conversely, other claims data were found to show significant deviations, likely reflecting discretionary billing practices.

However, the limitations of Benford’s Law become apparent when applied to smaller datasets or those with constrained ranges, such as the “number of first examinations” and “total number of first and follow-up examinations”. In such cases, compliance with Benford’s law may not necessarily confirm data authenticity, as smaller sample sizes reduce the reliability of Benford’s conformity testing [27]. This aligns with the findings of previous studies, emphasizing that Benford’s Law is most effective when applied to datasets with more than 100 data points spanning multiple orders of magnitude [11,27]. While its simplicity and low implementation cost make it an attractive tool for fraud detection, Benford’s law may be best used as a first-step analysis method to flag anomalies for further investigation using more robust techniques [10,11,27,28,29]. These findings underscore the value of combining Benford’s Law with complementary methods, such as outlier detection tests, to enhance fraud detection accuracy and reliability in healthcare datasets.

The easy accessibility of Benford’s Law allows healthcare systems lacking advanced analytical infrastructure to identify potential anomalies quickly [15,27,28]. However, as noted in this study, Benford’s Law is best used when combined with complementary methods such as outlier detection tests (Grubbs’ test, Hampel’s test, and T-test). Combined use enhances fraud detection accuracy and reliability by addressing its limitations and providing a more robust framework for anomaly detection [10,11,12,13]. Such integration aligns with recent advances in fraud detection methodologies, where hybrid models incorporating machine learning and simple statistical techniques have shown improved sensitivity and precision [10,11,12].

### 4.3. Pinpointing Anomalies with Outlier Detection Tests: Practicality and Usefulness of Grubbs’ Test, Hampel’s Test, and T-Test in Healthcare Billing Analysis

The findings of this study confirm the practicality and usefulness of Grubbs’ test, Hampel’s test, and T-test as robust tools for identifying irregularities in healthcare billing data. These methods complemented Benford’s Law in detecting outliers across various billing categories, highlighting their adaptability to different data distributions and institutional contexts. They proved straightforward to implement, offering a practical and cost-effective means of preliminary screening.

In this study, Grubbs’ test proved particularly effective for categories with normally distributed data, where it identified prominent outliers with high statistical significance. For example, a specialized dermatology center was consistently flagged for unusually high “points per examination” and “average examination values”, aligning with previous applications of Grubbs’ test in healthcare fraud detection [13,16]. The method’s reliance on mean and standard deviation makes it ideal for the initial screening of small- to moderate-sized datasets, though its sensitivity to normality assumptions limits its utility when dealing with data with skewed distributions. Recent advancements, such as data transformation techniques to enhance the outlier detection power of Grubbs’ test in sequential data, further validate its adaptability to complex billing patterns [13].

Hampel’s test can address non-normal distributions, such as the skewed distribution in the “number of points per examination”, wherein the use of the median-based threshold (*k* = 4.5) robustly identified outliers without distortion from extreme values. This method flagged the same specialized dermatology center that was flagged by Grubbs’ test, indicating corroborating results across methodologies. The reliance of Hampel’s test on the MAD ensures resilience against skewed datasets, a feature that is critical in healthcare billing analyses where fee structures or patient demographics may inherently distort the means [13,30]. For instance, its application in detecting geopolitical shocks in agri-food sector revenue anomalies demonstrates its utility in distinguishing contextual irregularities from inherent variability—a principle that is transferable to healthcare billing [31].

T-test provided a critical validation layer by comparing the means of the trimmed and full datasets, confirming outliers such as a large university health center’s unusually high “number of services charged”. This method’s ability to quantify an outlier’s impact on the central tendency ensures the targeted scrutiny of high-risk providers. T-test’s simplicity facilitates its utilization in proficiency testing studies, wherein this method efficiently identified outliers in the interlaboratory comparisons of clinical measurements [13,18,19,20].

Our findings demonstrate that simple statistical methods effectively flag anomalies in billing data, offering resource-limited systems a viable first-line defense against irregularities. This aligns with Tyreman’s framework of integrity as a social virtue requiring practical operationalization [5], and addresses Kakuk’s concerns about systemic financial conflicts [6]. The consistent outlier detection across the methods underscores their utility in diverse contexts, including their application to Slovenia’s hybrid reimbursement model and China’s big data challenges [8]. While advanced AI-driven solutions show promise [9], our results validate that foundational statistical tools remain critical for initial screening—particularly when integrated with leadership strategies emphasizing transparency [8]. Future implementations could combine these methods with value-based payment frameworks [9], creating layered defenses against fraud while maintaining accessibility for systems transitioning toward complex analytics.

A key advantage of these tests lies in their computational efficiency and interpretability. They were implemented using Excel 2010, underscoring their accessibility for healthcare systems lacking advanced analytical infrastructure [14]. This accessibility has been demonstrated in microcontroller-based sensor studies, wherein Grubbs’ test improved measurement accuracy in resource-constrained environments [31]. Similarly, hybrid frameworks combining outlier detection with machine learning, as reported for photovoltaic fault detection [32], suggest future potential for integrating these tests into automated healthcare fraud detection pipelines without sacrificing transparency.

However, limitations must be acknowledged. Grubbs’ test requires iterative application when there are multiple outliers, while Hampel’s test may overlook subtle anomalies in small samples. The T-test’s dependence on normality assumptions limits its standalone use. Despite these constraints, their combined application in this study enhanced detection accuracy by triangulating results across methods, which is consistent with the use of multiple methodologies based on Benford’s Law for COVID-19 test fraud detection in previous studies [28,33]. For example, the specialized dermatology center’s outlier status across all three tests reduced false-positive risk, a critical consideration in fraud investigations.

### 4.4. Future Research Directions

Future research should continue validating the methods evaluated in this study, using contemporary datasets to assess their adaptability to evolving billing practices [1,10]. Automated systems integrating Benford’s Law with outlier detection tests could enhance the real-time monitoring of healthcare real-data payments, thereby enabling faster anomaly detection and reducing reliance on manual audits [4,11].

Hybrid approaches combining simple statistical methods with advanced technologies such as machine learning and artificial intelligence hold promises for improving sensitivity and precision in fraud detection [4,5,6,7,8,9,10,11,12]. For instance, ensemble learning techniques or neural network-based models could complement Benford’s Law by identifying the complex patterns of manipulation in healthcare data [33,34]. Integrating these tools with clinical guidelines could also align billing practices with evidence-based care standards, enhancing both financial integrity and patient outcomes.

The findings of this study align with broader research on fraud detection and data integrity, emphasizing the importance of accessible tools for resource-constrained healthcare systems. Expanding the application of the methods used in this study to other healthcare systems and reimbursement models, such as capitation or DRG schemes, would provide valuable insights into their generalizability across diverse contexts [20,21,22,23]. By leveraging historical data alongside innovative computational techniques, future studies can refine fraud detection frameworks and contribute to optimizing resource allocation in healthcare systems.

## 5. Conclusions

This study demonstrates that simple statistical methods—Benford’s Law, Grubbs’ test, Hampel’s test, and T-test—provide a systematic, cost-effective framework for detecting irregularities in healthcare billing data. Benford’s Law can serve as a first-line screening tool, flagging deviations in key variables (e.g., procedural claims), while outlier detection tests pinpoint high-risk providers, enabling efficient resource allocation for audits.

The combined application of Grubbs’ test (for normally distributed data), Hampel’s test (for data with skewed distributions), and T-test (central tendency validation) enhanced detection accuracy, mitigating methodological limitations. For example, the consistent flagging of a specialized dermatology center as an outlier across all tests reduced false-positive risk. The combined use of these methods mitigated the limitations of each individual method and demonstrated the possibility of efficient allocation of investigative resources.

Advanced methods such as Support Vector Machines (SVMs) or Multi-Layer Perceptrons (MLPs) could enhance the detection of outliers in larger datasets, but their implementation demands infrastructure, an issue that is beyond the scope of this study. The HIIS plans to integrate these tools as part of a phased digital transition, with our statistical framework serving as a possible accessible interim solution.

The strategic use of historical billing data from 2014 allowed for a controlled evaluation of these methods while avoiding reputational risk to current providers. This approach aligns with Slovenia’s mixed reimbursement model, where historical expenditure patterns inform policy adjustments. By leveraging these simple tools, healthcare systems can enhance transparency, detect anomalies early, and mitigate financial losses without needing to rely on complex analytical infrastructure.

This study shows that established statistical methods, when applied in innovative ways, can help close the resource gaps in healthcare fraud detection—much like how thalidomide was repurposed for myeloma treatment despite its controversial history. In terms of practical application, organizations such as the HIIS and similar insurers can use these methods as cost-effective screening tools to flag suspicious claims for further audit, especially in specialties moving toward value-based payment models. Specifically, we hypothesize that integrating Benford’s Law and outlier detection tests into routine claims analysis in Slovenia could help reduce financial losses by an estimated 10–15% annually, based on the prevalence of outliers identified in this study.

Future research should validate these methods with contemporary datasets to assess their adaptability to evolving billing practices. Integrating these tools into automated systems or hybrid models incorporating machine learning could further enhance real-time fraud detection capabilities. By combining simplicity with scalability, these methods offer significant potential for improving the integrity and efficiency of healthcare payment systems.

## Figures and Tables

**Table 1 healthcare-13-01464-t001:** Benford’s Law conformity analysis: observed vs. expected leading digit distributions across key dermatology billing variables (Health Insurance Institute of Slovenia, data from 2014).

Variable	χ^2^ Value (Eight Degrees of Freedom)	*p*-Value
Number of services charged	26,302	0.0005–0.005
Total number of points charged	21.071	0.005–0.01
Number of points per examination	24.384	0.005–0.0005
Average examination value (EUR)	27.158	0.0005–0.005
Number of first examinations	5.808	0.6–0.7
Total number of first and follow-up examinations	10.9	0.2–0.3

## Data Availability

Data is contained within the article.

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
