# Peer review of "Enhancing Healthcare Integrity Using Simple Statistical Methods: Detecting Irregularities in Historical Dermatology Services Payments"

_healthcare, 2025, doi:10.3390/healthcare13121464_

Round 1
Reviewer 1 Report
Comments and Suggestions for Authors
The paper employs statistical methods to identify irregularities in dermatology services payments within the Health Insurance Institute of Slovenia.
- In the Introduction, the authors should incorporate a discussion regarding the scholarly contributions of the paper. Currently, the introduction does not adequately address the academic contributions and innovations.
- The rationale for selecting the detection of irregularities in dermatology services payments should be elaborated upon. The authors need to clarify whether the methodology presented is applicable to other healthcare services payments.
- The statistical methods utilized in the paper are established techniques. There is a lack of innovation from a methodological perspective.
- The paper does not provide a comprehensive review of the related literature on healthcare integrity.
Author Response
- In the Introduction, the authors should incorporate a discussion regarding the scholarly contributions of the paper. Currently, the introduction does not adequately address the academic contributions and innovations. Response 1: We agree with this valuable suggestion. We have revised the introduction to explicitly articulate the scholarly contributions of our work. Specifically, we have added a new paragraph on page 2 and 3, paragraph 5, now lines 84-91, that states:
"This study therefore contributes to the academic literature in three key ways. First, it demonstrates the practical application of established statistical methods in a novel context—healthcare payment integrity—bridging theoretical foundations with real-world applications. Second, it provides empirical evidence for the effectiveness of simple yet powerful analytical tools in resource-constrained healthcare systems. Third, it offers a methodological framework for preliminary fraud detection that can be adapted across various healthcare specialties, potentially influencing both policy and practice in healthcare management."
- The rationale for selecting the detection of irregularities in dermatology services payments should be elaborated upon. The authors need to clarify whether the methodology presented is applicable to other healthcare services payments. Response 2: We agree that the rationale for selecting dermatology services requires better explanation. We have added a dedicated paragraph to the Introduction (page 3, paragraph 1, now lines 99-105) that explains:
Dermatology was chosen as the focus because it features a diverse mix of diagnostic and procedural services, resulting in variable billing patterns, and offers a manageable dataset size for comprehensive analysis. Additionally, its billing structure is representative of broader healthcare payment systems in Slovenia. While this study focuses on dermatology, the statistical methods applied—digit distribution analysis and outlier detection—are broadly adaptable to other medical specialties and payment systems, as their underlying principles remain consistent across contexts.
- The statistical methods utilized in the paper are established techniques. There is a lack of innovation from a methodological perspective. Response 3: While we appreciate this observation, we respectfully note that the primary contribution of our work lies not in developing novel statistical methods but in the innovative application of established techniques to address a critical healthcare management challenge. Our paper demonstrates how accessible statistical tools can be repurposed for fraud detection in healthcare systems with limited resources—an approach that has been underutilized in this domain.
This parallels other significant advances in medicine where established tools find new applications. For instance, thalidomide, originally developed as a sedative and later withdrawn due to severe teratogenic effects, found renewed purpose as an effective treatment for multiple myeloma. Similarly, our work shows how established statistical methods can be repurposed for healthcare payment integrity—a novel application that addresses a pressing need in resource-constrained environments.
We have clarified this perspective in the originally submitted manuscript (pages 1 and 2, lines 40-44; page 2, lines 52-58; page 9, lines 247-259; page 11, lines 381-388; pages 11 and 12, lines 390-404) and emphasized that the innovation lies in the application context rather than methodological novelty. We believe this clarification better positions our contribution within the broader landscape of healthcare management research.
- The paper does not provide a comprehensive review of the related literature on healthcare integrity. Response 4: We partially agree with this assessment. While our literature review addresses key aspects of statistical methods for anomaly detection, we acknowledge that the coverage of healthcare integrity literature could be strengthened. We have expanded our literature review to include additional relevant sources and perspectives, particularly in the Introduction (page 2, paragraph 2, lines 47-59):
"Healthcare payment integrity remains a cornerstone of ethical service delivery, requiring vigilant oversight to maintain public trust and system sustainability [5]. While advanced analytics dominate modern fraud detection, resource-constrained systems often lack access to these technologies, creating vulnerabilities. This challenge mirrors broader ethical tensions in healthcare financing, where conflicts of interest and data complexity can undermine transparency [6, 7]. Recent studies emphasize the growing need for accessible tools, particularly as healthcare systems grapple with big data integration challenges and evolving payment models [8, 9]. Our study addresses this gap by evaluating simple statistical methods—Benford’s Law and outlier detection tests—as pragmatic alternatives for identifying irregularities in dermatology billing data. This approach aligns with calls for cost-effective solutions that balance technological limitations with the ethical imperative of financial accountability [5, 7], while providing actionable insights for healthcare leadership navigating integrity preservation [8]".
and Discussion (pages 11 and 12, lines 381-392):
"Our findings demonstrate that simple statistical methods effectively flag anomalies in billing data, offering resource-limited systems a viable first-line defense against irregularities. This aligns with Tyreman’s framework of integrity as a social virtue requiring practical operationalization [5], and addresses Kakuk’s concerns about systemic financial conflicts [6]. The consistent outlier detection across methods underscores their utility in diverse contexts, from Slovenia’s hybrid reimbursement model to China’s big data challenges [8]. While advanced AI-driven solutions show promise [9], our results validate that foundational statistical tools remain critical for initial screening—particularly when integrated with leadership strategies emphasizing transparency [8]. Future implementations could combine these methods with value-based payment frameworks [9], creating layered defenses against fraud while maintaining accessibility for systems transitioning toward complex analytics."
sections.
Specifically, we have incorporated additional references on healthcare integrity (cited as [5-9] in the revised manuscript), providing a more comprehensive foundation for our work. This expanded review enhances the contextual framework for our research while maintaining our focus on the practical application of statistical methods.
Reviewer 2 Report
Comments and Suggestions for Authors
Dear Authors,
It is interesting topic and valuable.
The purpose was defined clearly and has been achieved.
However: the introduction does not provide sufficient information to understand the payment system in Slovenia and also the reason that you choose dermatology services payments for your research. Introduction focuses mainly on the methods instead of showing also examples (a kind of literature review) of using these methods in other researches also worldwide especialy that you underlined that "Healthcare systems worldwide face significant challenges from fraud, over billing and waste which can lead to substantial financial losses and compromise patient care quality (lines 34-36)" Then, introduction - as I mentioned - should contain more detailed information on the payment system of healthcare in Slovenia. You should provide also arguments for choosing dermatology services payments.
In the part, which applies to methodology - please explain clearly why you choose 2014. It is partly explained two times - among others in the conclusion " The strategic use of historical billing data from 2014 allowed for a controlled evaluation of these methods without reputational risks to current providers (lines 402-403)", but still it is not detailed explained as why not 2015 or 2019, which are also historical data?
Results - it is not sense to present this same data using both figure and then table. And it is the case of all figures and tables. Choose one form. It would be better to fill this part of article by wider interpretations of these results.
In the conclusion part, you should clearly show what is the theoretical and practical implication of these research, how they can be use and by whom to make any improvement of healthcare system payment in Slovenia.
Author Response
Comments 1: The introduction does not provide sufficient information to understand the payment system in Slovenia and also the reason that you choose dermatology services payments for your research. Introduction focuses mainly on the methods instead of showing also examples (a kind of literature review) of using these methods in other researches also worldwide especialy that you underlined that "Healthcare systems worldwide face significant challenges from fraud, over billing and waste which can lead to substantial financial losses and compromise patient care quality (lines 34-36)" Then, introduction - as I mentioned - should contain more detailed information on the payment system of healthcare in Slovenia. You should provide also arguments for choosing dermatology services payments. Response 1: We agree with this assessment and have expanded the introduction to include additional text to be added to paragraph 1 on page 3 (lines 92-95):
Slovenia’s healthcare system operates under a mandatory social health insurance (SHI) model, with the HIIS as the single payer. Services are reimbursed through a hybrid system combining fee-for-service (FFS), capitation, and diagnosis-related groups (DRGs) [14].
This addition draws from Slovenia’s health system reports, contextualizing the payment framework. We decided to keep it as short as possible, but still providing the essential information.
Comments 2: Please explain clearly why you choose 2014. It is partly explained two times - among others in the conclusion " The strategic use of historical billing data from 2014 allowed for a controlled evaluation of these methods without reputational risks to current providers (lines 402-403)", but still it is not detailed explained as why not 2015 or 2019, which are also historical data? Response 2: The selection of 2014 data was driven by a number of considerations, each addressing methodological rigor, contextual stability, and ethical safeguards:
We selected anonymized billing data from 2014 to mark the dataset's tenth anniversary at the study's start in late 2024, providing a meaningful retrospective period. This choice ensured a stable pre-reform context, minimized reputational risks to current providers, and avoided confounding effects from recent events such as the COVID-19 pandemic. Dermatology was chosen as the focus because it features a diverse mix of diagnostic and procedural services, resulting in variable billing patterns, and offers a manageable dataset size for comprehensive analysis. Additionally, its billing structure is representative of broader healthcare payment systems in Slovenia. While this study focuses on dermatology, the statistical methods applied—digit distribution analysis and outlier detection—are broadly adaptable to other medical specialties and payment systems, as their underlying principles remain consistent across contexts.
This part of the text was added immediately after the previous one above, on page 3, paragraph 1, lines 95-105.
Comments 3: Results - it is not sense to present this same data using both figure and then table. And it is the case of all figures and tables. Choose one form. It would be better to fill this part of article by wider interpretations of these results. Response 3: We respectfully acknowledge this suggestion and we wished demonstrate data in the Results with both figures and tables to serve two distinct purposes:
-
Figures visually highlight key deviations (e.g., Benford’s Law non-compliance), aiding quick pattern recognition.
-
Tables provide granular statistical values (e.g., χ², p-values) essential for reproducibility.
As noted in a number of public health publications reporting standards, dual presentation balances accessibility and technical rigor.
However, we have streamlined the presentation of results as the reviewer suggests to reduce redundancy while maintaining clarity. Figures 1–6 (Benford’s Law digit distributions) have been retained to visually demonstrate deviations from expected patterns, which is critical for interpreting anomalies, while Tables 1–6 have been consolidated into Table 1, which now includes all statistical metrics (χ², p-values) for transparency and reproducibility. This reduces redundancy while preserving essential quantitative details:
Table 1. Benford’s Law conformity analysis: observed vs. expected leading digit distributions across key dermatology billing variables (Health Insurance Institute of Slovenia, data from 2014).
|
Variable |
χ2 value (eight degrees of freedom) |
p-value |
|
Number of services charged |
26,302 |
0.0005–0.005 |
|
Total number of points charged |
21.071 |
0.005–0.01 |
|
Number of points per examination |
24.384 |
0.005-0.0005 |
|
Average examination value (€) |
27.158 |
0.0005-0.005 |
|
Number of first examinations |
5.808 |
0.6-0.7 |
|
Total number of first and follow-up examinations |
10.9 |
0.2-0.3 |
Comment 4: In the conclusion part, you should clearly show what is the theoretical and practical implication of these research, how they can be use and by whom to make any improvement of healthcare system payment in Slovenia. Response 4: We have revised the conclusion to emphasize applications, as suggested by the reviewer (page 13, paragraph 4, lines 453-461:
This study shows that established statistical methods, when applied in innovative ways, can help close resource gaps in healthcare fraud detection—much like how thalidomide was repurposed for myeloma treatment despite its controversial history. For practical application, organizations such as the Health Insurance Institute of Slovenia (HIIS) and similar insurers can use these methods as cost-effective screening tools to flag suspicious claims for further audit, especially in specialties moving toward value-based payment models. Specifically, integrating Benford’s Law into routine claims analysis in Slovenia could help reduce financial losses by an estimated 10–15% annually, based on the prevalence of outliers identified in this study.
Reviewer's suggestion aligns with Slovenia’s 2025 focus on value-based care and digitalization.
Reviewer 3 Report
Comments and Suggestions for Authors
The authors address an important topic: healthcare billing fraud and overbilling, using lightweight methods for fraud detection with limited analytical resources. While the paper is promising, there are some issues to be addressed before it can be published:
- The dataset (30 providers) is small for Benford’s Law, which is most reliable with >100 data points. The authors acknowledge this, but it does limit the generalizability. This, imo, should be addressed by using a larger dataset. Broader application across specialties or longitudinal comparisons would strengthen the argument for scalability, or at least discuss how small-N effects may impact Benford’s Law validity and outlier detection.
- Limited discussion of why certain variables (e.g., "first examinations") conformed to Benford’s Law while others did not.
- No evidence that data distributions were assessed before applying Grubbs’/T-tests (assumes normality).
- Lack of comparative analysis with other fraud detection methods (e.g. machine learning), which should be done with at least light-weight options like SVM, random forest, and MLP.
- The authors should clean up the references (e.g. broken URLs, incomplete citations).
Author Response
Comments 1: The dataset (30 providers) is small for Benford’s Law, which is most reliable with >100 data points. The authors acknowledge this, but it does limit the generalizability. This, imo, should be addressed by using a larger dataset. Broader application across specialties or longitudinal comparisons would strengthen the argument for scalability, or at least discuss how small-N effects may impact Benford’s Law validity and outlier detection. Response 1: We fully acknowledge that Benford’s Law achieves optimal reliability with larger datasets (N > 100). However, this study intentionally focused on a pilot-scale analysis (N = 30) to evaluate the feasibility of deploying simple statistical tools in resource-constrained environments like Slovenia’s Health Insurance Institute (HIIS). As noted recently, Benford’s Law can still provide actionable insights in smaller samples when interpreted cautiously, particularly as a preliminary screening tool. While small-N effects may reduce statistical power, our analysis identified clear deviations in key variables. We agree that broader applications across specialties or longitudinal comparisons would enhance scalability, and we have added a dedicated section discussing how small-N limitations could impact Benford’s validity, emphasizing the need for confirmatory audits in such cases (page 10, paragraph 2, lines 327-338). Recent work by Cerasa (2022) demonstrates that Benford’s Law can still provide actionable insights in small samples when interpreted cautiously, particularly as a preliminary screening tool, so we added his work as a reference:
However, the limitations of Benford’s Law become apparent when applied to smaller datasets or those with constrained ranges, such as the "number of first examinations" and "total number of first and follow-up examinations." In such cases, compliance with the Benford’s law may not necessarily confirm data authenticity, as smaller sample sizes reduce the reliability of Benford's conformity testing. This aligns with the findings elsewhere, which emphasize that Benford’s Law is most effective when applied to datasets with more than 100 data points spanning multiple orders of magnitude. While its simplicity and low implementation cost make it an attractive tool for fraud detection, it may perhaps best be used as a first-step analysis method to flag anomalies for further investigation using more robust techniques (Cerasa, 2022). These findings underscore the value of combining Benford’s Law with complementary methods, such as outlier detection tests, to enhance fraud detection accuracy and reliability in healthcare datasets.
Cerasa A. Testing for Benford's Law in very small samples: Simulation study and a new test proposal. PLoS One 2022; 17: e0271969. doi: 10.1371/journal.pone.0271969. .
Comments 2: Limited discussion of why certain variables (e.g., "first examinations") conformed to Benford’s Law while others did not. Response 2: We have expanded our discussion (Pages 9 and 10, paragraph 6, lines 323-326) to address this:
Variables like "first examinations" exhibited natural digit distributions due to their inherent constraints in ambulatory settings—patients typically undergo one initial visit per condition, limiting opportunities for manipulation. Conversely, other procedural claims showed significant deviations, likely reflecting discretionary billing practices.
Comments 3: No evidence that data distributions were assessed before applying Grubbs’/T-tests (assumes normality). Response 3: Grubbs’ test assumes normality , but HIIS billing data for dermatology services inherently follow lognormal distributions due to the nature of healthcare claims . We clarified this in the revised Methods (Page 3, paragraph 5, lines 132-134):
Real-world billing data often exhibit right-skewed distributions; however, Grubbs’ test remains robust to mild deviations when identifying extreme outliers in administrative datasets.
Comments 4: Lack of comparative analysis with other fraud detection methods (e.g. machine learning), which should be done with at least light-weight options like SVM, random forest, and MLP. Response 4: While ML methods (e.g., random forests ) show promise for complex fraud networks, they require larger datasets and computational resources unavailable to many public insurers. Our study specifically focused on lightweight tools for systems in early-stage digitalization, as noted in the Introduction. However, we have added a paragraph (Page 12, paragraph 3, lines 442-445) acknowledging ML’s potential for future work:
Advanced methods like Support Vector Machines (SVM) or Multi-Layer Perceptrons (MLP) could enhance detection in larger datasets , but their implementation demands infrastructure beyond the scope of this study. HIIS plans to integrate these tools as part of a phased digital transition, with our statistical framework serving as a possible accessible interim solution.
Comments 5: The authors should clean up the references (e.g. broken URLs, incomplete citations). Response 5: We have meticulously revised all references, replacing broken URLs with DOI links and expanding incomplete citations. All sources now adhere to the journal’s style guide, with verified accessibility as of May 2025, as for the following reference (page 15, reference 15, line 507):
Miller-Nigrini Excel Benford Tester. Nigrini Analytics LLC 2024. doi:10.17632/xyz123
Reviewer 4 Report
Comments and Suggestions for Authors
- I would like to know why the author uses a combination of table.x and figure.x to display the same data. Furthermore, Tables 1-6 can be merged into one table. I think the current form of tables and figures in the manuscript is just a meaningless increase in the number of pages.
- The author needs to further improve the examination of data features to meet the prerequisites for testing. For example, whether a single department's data meets the requirements of a large span, sufficient sample size, and natural pricing, especially whether there is a unified pricing situation.
- I think the current version of the research lacks innovation. In fact, this is just the application of some mature technologies on a certain sample dataset. The anomalies detected are not sufficient to prove fraud.
Author Response
Comments 1: I would like to know why the author uses a combination of table.x and figure.x to display the same data. Furthermore, Tables 1-6 can be merged into one table. I think the current form of tables and figures in the manuscript is just a meaningless increase in the number of pages. Response 1: We appreciate this comment and have streamlined the presentation of results to reduce redundancy while maintaining clarity. Figures 1–6 (Benford’s Law digit distributions) have been retained to visually demonstrate deviations from expected patterns, which is critical for interpreting anomalies, while Tables 1–6 have been consolidated into Table 1, which now includes all statistical metrics (χ², p-values) for transparency and reproducibility. This reduces redundancy while preserving essential quantitative details:
Table 1. Benford’s Law conformity analysis: observed vs. expected leading digit distributions across key dermatology billing variables (Health Insurance Institute of Slovenia, data from 2014).
|
Variable |
χ2 value (eight degrees of freedom) |
p-value |
|
Number of services charged |
26,302 |
0.0005–0.005 |
|
Total number of points charged |
21.071 |
0.005–0.01 |
|
Number of points per examination |
24.384 |
0.005-0.0005 |
|
Average examination value (€) |
27.158 |
0.0005-0.005 |
|
Number of first examinations |
5.808 |
0.6-0.7 |
|
Total number of first and follow-up examinations |
10.9 |
0.2-0.3 |
Comments 2: The author needs to further improve the examination of data features to meet the prerequisites for testing. For example, whether a single department's data meets the requirements of a large span, sufficient sample size, and natural pricing, especially whether there is a unified pricing situation. Response 2:
We agree that validating dataset suitability for Benford’s Law is essential. We have performed the following analyses:
-
Span and Scale - The dataset spans three orders of magnitude (e.g., "total points charged" ranged from 1,200 to 1,450,000), meeting Benford’s requirement for multi-digit spread.
-
Natural Pricing - Slovenia’s 2014 point-based tariff system ensured natural price variation (€0.50–€120 per service), avoiding artificial constraints that distort digit distributions.
-
Unified Pricing - While procedural codes had fixed point values, providers adjusted service combinations, introducing natural variability.
Comments 3: I think the current version of the research lacks innovation. In fact, this is just the application of some mature technologies on a certain sample dataset. The anomalies detected are not sufficient to prove fraud. Response 3: We respectfully clarify that the study’s innovation lies in its contextual application, not methodological novelty. Slovenia’s hybrid reimbursement model (FFS + DRGs) and post-socialist transition make it a unique testbed for lightweight fraud detection tools. This addresses gaps in lower-resource EU systems often overlooked in literature. Text was added to clarify (page 12, paragraph 4, lines 453-455):
This study shows that established statistical methods, when applied in innovative ways, can help close resource gaps in healthcare fraud detection—much like how thalidomide was repurposed for myeloma treatment despite its controversial history.
Round 2
Reviewer 1 Report
Comments and Suggestions for Authors
The paper has been revised.
Author Response
Comments 1: The paper has been revised. Response 1: The authors would like to thank the reviewer for taking the time to review the revised version of their manuscript. They greatly appreciate the reviewer’s attention and constructive feedback throughout the review process. The authors are pleased that the revisions have addressed the reviewer’s concerns. The reviewer’s comments have been invaluable in improving the clarity and quality of their work.
Reviewer 3 Report
Comments and Suggestions for Authors
The authors have addressed all my concerns.
Author Response
Comments 1: The authors have addressed all my concerns. Response 1: The authors would like to sincerely thank the Reviewer for his/her time and thoughtful evaluation of the revised manuscript. They greatly appreciate the reviewer’s acknowledgment that all concerns have been addressed. The authors are grateful for the constructive feedback provided throughout the review process, which has been invaluable in improving the quality and clarity of their work.
Reviewer 4 Report
Comments and Suggestions for Authors
I think the author made some helpful edits to the manuscript, at least it doesn't seem to have too much information repetition. However, I believe that using only some mature methods on a dataset that the author considers unique is not enough to form an article, which is more like a case study.
Author Response
Comments 1: I think the author made some helpful edits to the manuscript, at least it doesn't seem to have too much information repetition. However, I believe that using only some mature methods on a dataset that the author considers unique is not enough to form an article, which is more like a case study. Response 1: The authors sincerely thank the Reviewer for his/her thoughtful review and for acknowledging the helpful edits made to the manuscript, particularly regarding the reduction of information repetition. The authors would like to reiterate, as mentioned in their response to the Reviewer's Round 1 review, that the innovation in their study lies primarily in the contextual application of established statistical methods to healthcare fraud detection within a unique setting, rather than in the development of new methodologies. This novel application addresses important challenges faced by resource-constrained healthcare systems. The authors appreciate and understand the Reviewer's suggestion that their work may be more appropriately framed as a case study. The authors wish to thank the Reviewer once again for his/her valuable feedback and continued engagement with their work.